# COACH CV: The Seven Clinical Phenotypes of Concussion

**DOI:** 10.3390/brainsci7090119

**Published:** 2017-09-16

**Authors:** Neil Craton, Haitham Ali, Stephane Lenoski

**Affiliations:** 1Faculty of Medicine, University of Manitoba, 14-160 Meadowood Drive, Winnipeg, MB R2M 5L6, Canada; dr.haithamali@hotmail.com (H.A.); umlenoss@myumanitoba.ca (S.L.); 2School of Kinesiology and Applied Health, University of Winnipeg, Winnipeg, MB R3B 2E9, Canada

**Keywords:** concussion, assessment, phenotype, trajectory, clinical, physical examination, whiplash

## Abstract

Our understanding of the diverse physiological manifestations of concussion is changing rapidly. This has an influence on the clinical assessment of patients who have sustained a concussion. The 2017 Consensus Statement on Concussion in Sport states that numerous post-injury clinical findings, such as cognitive deficits, post-traumatic headaches, dizziness, difficulties with oculomotor function, and depression have all been associated with a poorer prognosis in concussed patients. This demonstrates that there are several potential clinical manifestations after head injury warranting clinical evaluation. We have developed an acronym to guide the office-based assessment of concussed patients to consider each of the potential clinical phenotypes. “COACH CV” prompts the clinician to evaluate for cognitive problems, oculomotor dysfunction, affective disturbances, cervical spine disorders, headaches, and cardiovascular and vestibular anomalies.

The Sport Concussion Assessment Tool, fifth edition (SCAT5) is recommended for the evaluation of the concussed patient on the sidelines. It documents symptoms of concussion, orientation, memory, cervical spine tenderness, balance, and coordination [1]. It does not include detailed oculomotor screening, vestibular testing or cardiovascular assessment, which may be too cumbersome for the sideline context. For the office-based clinical assessment, the Berlin Guidelines recommend an assessment that includes a comprehensive history and detailed neurological examination including a thorough assessment of mental status, cognitive functioning, sleep/wake disturbance, ocular function, vestibular function, gait and balance [1]. The diagnostic utility of the SCAT5 decreases in the first 3–5 days, the timeframe in which many concussed patients will be evaluated by their primary health care provider [1]. Primary health care providers may not be familiar with the diverse manifestations of concussion. Several publications have documented the considerable overlap between concussion, whiplash and vestibular disorders [2,3,4,5]. Oculomotor and cardiovascular anomalies are well-documented after concussion [5,6,7]. In addition, pre-accident psychological disturbances worsen prognosis, and the potential for affective disturbances has been shown in those suffering persisting symptoms after concussion [8]. These conditions should be considered in the office-based assessment of concussed patients to gauge prognosis and to guide treatment [1]. Evaluating these different phenotypes is an important part of the clinical assessment of head-injured patients, as the most recent iteration of concussion guidelines has included a significant exclusion clause. The Berlin document has stated that concussion should not be diagnosed if another condition can explain the patient’s clinical findings. It has stated in the definition of concussion that the patient’s clinical signs and symptoms cannot be explained by cervical injuries, vestibular dysfunction, psychological factors, medication use or coexisting medical conditions [1]. The clinical phenotypes overlap to a significant degree, and they are not meant to be evaluated in isolation. For example, a symptom such as dizziness can be caused by cervical dysfunction, oculomotor difficulties, and vestibular and cardiovascular disorders after concussion. The primary care provider is encouraged to consider each potential phenotype in a patient with delayed recovery.

## 1. C Is for Cognitive Function

Cognitive difficulties such as memory impairment, decreased attention, impaired concentration, slowed mental processing speed and other areas of executive dysfunction are common in concussed patients, especially in the acute recovery period [1]. The identification of these symptoms is central to the assessment of concussion. There are several methods to document cognitive function. The SCAT5 can be administered immediately after a concussion and can be used on the sidelines or in the locker room. It can be repeated throughout the recovery period, but its diagnostic utility decreases in the first 3–5 days [1]. In addition to the Glasgow Coma Scale (GCS) and orientation, it documents patient symptoms and provides a symptom score using a 22 item scale. It also documents immediate and delayed memory, concentration and tasks that assess processing speed and attention, such as reciting the months of the year backward. Computer-based neuropsychological tests such as ImPACT and Cogsport also document symptoms related to cognitive dysfunction and assess visual and verbal memory, reaction time and the speed of information processing. Confounding factors that can influence individual performance on cognitive tests, such as mood, test location, headache, pain, fatigue, anxiety, and –Attention Deficit Hyperactivity Disorder should be noted [9]. It should be acknowledged that cognitive symptoms are not specific to concussion [1]. Healthy people have concussion symptoms -most of the time, and patients involved in traumatic episodes with no head injury also report cognitive difficulties and impairment [2,10]. When cognitive symptoms persist beyond two weeks, referral to a neuropsychologist for the formal evaluation of cognitive and emotional function should be considered [1].

## 2. O Is for Oculomotor Manifestations

Several oculomotor manifestations have been associated with concussion [11,12]. Abnormalities in the near point of convergence (NPC), saccadic eye movements, visual pursuit and screen viewing are common, and as is photophobia; up to 61% of patients report symptom provocation after oculomotor screening [11,12,13,14]. Examining patients for these problems is important, as oculomotor disturbances have been associated with protracted recovery from concussion [15]. Pearce et al. showed that convergence insufficiency (CI) was reported in 42% of their concussed patients [13]. Although CI can exist without concussion, they concluded that CI is a poor prognostic indicator and should be evaluated to guide treatment, academic recommendations and potential referral for optometric assessment [5,13,15]. In addition, CI has been shown to be associated with other concussion symptoms, such as blurred vision, difficulty focusing and headaches [15]. Abnormal saccadic eye movements are also associated with concussion [12]. They have been associated with impaired cognitive function and chronic post-concussion symptom reporting [12]. Assessment modalities such as the King–Devick (K–D) test, used by some concussion specialists, can be used to evaluate saccadic eye movements in an objective manner and can supplement a standard clinical assessment [16]. Other methods, such as goggles to enhance the appearance of abnormal saccades and saccadometric lasers, are available. Abnormalities in smooth visual pursuit can also be found in a concussed patient [12]. Smooth visual pursuit reflects the function of the ocular-cerebellar network in addition to the cognitive functions of attention and working memory [17]. We would suggest that screening for oculomotor dysfunction should be part of the routine office-based assessment of concussed patients. Referral to an optometrist with experience in assessing concussed patients for formal evaluation is a reasonable consideration.

## 3. A Is for Affective Disturbances

The primary care practitioner needs to be aware that affective disturbances are a risk factor for persisting symptoms after concussion, and that they are potential consequences of concussion [1]. There is a considerable overlap between the symptoms of depression and concussion. Affective symptoms are part of the clinical expression of concussion and are embedded in the concussion symptom scale (fatigue, sadness, irritability, trouble sleeping, difficulty concentrating and being emotional or anxious) [1]. Pre-accident affective disturbances are a risk factor for persisting symptoms after concussion, and there is a growing body of literature indicating that psychological factors play a significant role in delayed recovery [1,18,19]. Establishing the patient’s history of psychological disturbances is an important part in the office-based assessment of a concussed patient. Studies on pediatric sport-related concussion have shown that affective symptoms were common in pediatric concussion patients [20]. A retrospective study evaluating retired National Football League players showed an increase in depressive symptoms in the players who had sustained a higher number of concussions [21]. Several publications have stressed the need for future prospective studies that use standardized assessment tools to document the prevalence and risk factors of post-concussion affective problems, including depression, anxiety and bipolar disorders [18]. Inventories such as the Patient Health Questionnaire 9 (PHQ-9) lend themselves to identifying depression in concussed patients, but they have not been validated in this population. Most of the symptoms identified in the PHQ-9 are found on the concussion symptom scale included in the SCAT5. Although controversial, some studies have shown an increased risk of suicide in patients who have sustained a concussion [22]. This is contrasted with a recent systematic review that concluded there was little evidence of long-term psychological manifestations after concussion [8]. It is unknown whether concussion increases the risk for suicide in any meaningful way. We advocate a high degree of clinical vigilance for affective disturbances in the concussed patient, and suggest documenting the patient’s history of mood disorders as well as the potential for suicidal ideation, with careful attention to the depressive symptoms embedded in the SCAT5 symptom scale. Clinicians can treat depression in the concussed patient with the usual modalities of counselling, chemotherapy and psychological/psychiatric referral as required.

## 4. C Is for Cervical

It is important for clinicians to recognize that there is a significant overlap between concussion and whiplash [2]. The mechanism of injury in concussion and in whiplash is essentially the same, and the conditions have an identical symptom expression. The prevalence of cervical spine pathology in concussed patients is unknown [4] Cranio-cervical perturbation can affect the cervical nerve roots, cervico-thoracic and cervico-scapular musculature, and the trigemino-cervical nucleus, as well as the cervical inter-vertebral discs and zygapophyseal joints. These structures can contribute to neck pain, headaches, dizziness and balance difficulties [23,24,25,26,27]. Matuszak reviewed the physical assessment of concussed individuals and concluded that examining the cervical spine, assessing the neck range of motion, conducting the Spurling test and palpating the muscular and bony anatomy of the neck were important [28]. In a more recent systematic review, Cheever et al. highlighted the overlap between concussion and whiplash injury [4]. They advocated a comprehensive cervical spine assessment in the concussed patient, including the cervical joint-reposition error test, the smooth pursuit neck torsion test, the head–neck differentiation test, and the cervical flexion–rotation test, as well as cervical spine palpation. The relevance of cervical spine assessment was supported in the systematic review conducted by Marshall et al. and in their short case series, showing the benefits of therapy for the cervical spine in concussed individuals [29]. Schneider also demonstrated that physical therapy for the cervical spine and vestibular system could help to expedite the recovery of concussed adolescents [3]. The cervical spine is a potential source of multiple symptoms in the concussed individual and should be carefully evaluated after excluding more serious cervical conditions using clinical algorithms such as the Canadian C-spine rules [30]. While most primary care physicians may lack the manual skills required to elucidate the cervical manifestations generating concussion symptoms, allied health practitioners such as physiotherapists can assist in this regard [3,4]. There is evidence that physical therapy for the cervical spine can help patients recover from their injury more quickly than a control group [1,3].

## 5. H Is for Headaches

All practitioners should be aware that several different types of headache can follow concussion. Headache is the most common symptom in concussed patients [1]. Migraine headache, tension headache, cervico-genic headache and post-traumatic headache are all observed in the concussed-patient population [31,32,33]. Red flags for the patient with post traumatic headache include a deteriorating level of consciousness, progressive worsening, vomiting, early morning headache and headache worsening in the supine position. Patients should be evaluated according to an algorithm such as the Canadian Computed Tomography (CT) head injury rules for the requirement of advanced imaging. In a recent case-control observational study, Eckner showed an association between migraine headaches and concussion, concluding that pre-existing migraine headaches are a risk factor for chronic headache reporting after concussion [31]. Seifert cautioned that headaches could represent an exacerbation of pre-concussion headaches, rather than a new brain injury [32]. The treatment of post-traumatic migraine or post-traumatic tension headaches was considered important. As previously noted, evaluating for cervico-genic headache is an evidence-based step in the assessment and treatment of concussed patients [3,4]. Physiotherapy has proven to be beneficial for this combination of phenotypes [1,3,4]. Minen concluded that there was an association between headache and potential complications of concussion, including depression, insomnia, anxiety and cognitive difficulties, stressing the importance of assessing and treating this phenotype [33]. Standard analgesics and anti-migraine treatments can be used, but the role of pharmacotherapy for concussion symptoms is not well established [1].

## 6. C Is for Cardiovascular

An interesting fact that seems unknown by many primary care practitioners is that several cardiovascular manifestations are associated with concussion. These manifestations include exercise intolerance, altered heart rate variability, postural orthostatic tachycardia syndrome (POTS), autonomic nervous system anomalies, elevated heart rate and others [7,35]. Several studies have documented an “uncoupling” of the autonomic nervous system and the cardiovascular system with abnormal heart rate in the concussed athlete [34,35]. This is often referred to as dysautonomia. In their review, Matuszak et al. recommended including orthostatic vital signs in the clinical assessment of concussed patients if dizziness or imbalance were present [28]. Heyer et al. demonstrated that POTS was common in patients with persisting lightheadedness and vertigo after concussion [7]. In their sample, 71% of patients with lightheadedness or vertigo after concussion had POTS [7]. POTS was correlated with symptoms, and all patients with resolution of POTS had corresponding improvements in their symptoms, including lightheadedness and vertigo [7]. Although not routinely performed and not usually required, both exercise stress testing and tilt table testing can reveal abnormal cardiovascular function in patients who have persisting post-concussion symptoms, despite normal orthostatic vital signs [1,7]. Measuring heart rate, blood pressure and determining orthostatic changes are objective measures that should be part of a routine concussion assessment, and they are well within the competency of primary care practitioners.

## 7. V Is for Vestibular

Dysfunction of the vestibular system is another common problem following a concussion; up to 81% of patients show a vestibular abnormality on initial clinical examination [5,36]. The vestibular system receives multiple inputs from several other neuromuscular systems, including the oculomotor pathways, the brainstem, spinal cord, cervical spine, cerebral cortex, and cerebellar and peripheral sensory systems. Concussed patients can present with dizziness, vertigo, balance problems and gait difficulties, as well as the visual symptoms (covered in the oculomotor section) [5,11]. These manifestations are associated with a higher risk of persisting symptoms after concussion. Vestibular physiotherapy has been shown to accelerate recovery with a more expeditious return to sport in a group of concussed pediatric athletes, and it has been endorsed by the most recent consensus statement on sport-related concussion [1,3]. Therefore, screening for vestibular symptoms and signs should be part of the routine office-based assessment of concussed patients. Specific clinical examination of the vestibular system should include the following: the Romberg test, tandem gait, the Dix–Hallpike maneuver, and the vestibulo-ocular reflex (as well as the oculomotor screen described above) [5,37]. Those working in concussion clinics will often use the Balance Error Scoring System (BESS) to document vestibulo-spinal anomalies. Vestibular impairments can occur without balance and saccadic abnormalities [37]. This has prompted the development of the Vestibular/Ocular Motor Screening (VOMS) tool, which includes smooth pursuit, horizontal and vertical saccades, NPC, horizontal and vertical Vestibulo-ocular reflex (VOR), and visual motion sensitivity. The VOMS tool has been evaluated in concussed adolescent athletes by comparing the results of a cross-sectional design with those of the K–D test and the BESS. The screening tool identified vestibular dysfunctions other than those measured by the BESS or K–D alone [16,37]. This complete VOMS assessment assists in the accurate and efficient identification of anomalies in the vestibular system [5]. A complete neurological exam consisting of cranial nerve, motor/sensory/reflex and cerebellar findings should be included to rule out any neurological deficits.

## 8. Conclusions

The understanding of the physiological manifestations of concussion continues to evolve. The primary care clinician is faced with a wide array of potential signs and symptoms related to this disorder, which may be associated with persisting symptoms. We have developed an acronym to guide the office-based assessment of concussed patients that evaluates each potential trajectory of this disorder. COACH CV can help clinicians to consider the different phenotypes of concussion in an effort to ensure that the clinical manifestations are identified and treated.

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
