# Peer review of "COACH CV: The Seven Clinical Phenotypes of Concussion"

_brainsci, 2017, doi:10.3390/brainsci7090119_

Round 1

Reviewer 1 Report

The authors have removed the inappropriate content from the prior submission. The paper has some room for improvement in clarity and citations but I think will make an acceptable contribution to the literature if they can address a few remaining comments.

The paper provides a concise overview of different clinical
characteristics that be involved in post-concussion syndrome. The paper covers
an appropriate scope of information, cites current literature, and could be
useful for primary care providers who desire a brief overview of things to
consider in assessing patients with post-concussive symptoms. The authors rely
too much on citing secondary sources and there are a couple areas for
improvement in clarity of the report, as detailed below.

--2nd sentence (line 27-28): orientation and memory are part
of the cognitive assessment – just list ’cognitive function’ or the component
parts of the cognitive assessment so as not to be redundant

--line 33 missing right parenthesis

--line 39 – ‘potential for suicide has been shown in those suffering
persisting symptoms after concussion’ – this is overly alarmist and contradicts
results of large-scale studies finding lower suicide rates in former NFL
players. Consider removing.

--line 57 – unclear what is being referenced by “information processing
tasks” – your other statements list the major components of the SAC so this may
not be needed

--line 96 – missing period

--line 106 – I’m not a big fan of the suicide topic here. Although you
acknowledge that this is controversial and cite contradictory evidence, I think
people in this space believe that sustaining a concussion causally increases
suicide risk in any meaningful way. This kind of research requires careful
attention to candidate third/confounding variables and I’m not sure the article
you cite adequately did this. If this were true shouldn’t we see a higher incidence
of suicide in former NFL players? Monitoring suicide risk in patients with
depression is important in its own right, but I think in trying to draw a link
between concussions and suicide, you risk losing your audience.

--135—unless the journal allows it due to limits in the number of
citations, you should be citing primary sources when mentioning research
findings and rely on reviews for mentioning broad conclusions or expert consensus
statements specifically mentioned in those types of papers (same statement
applies elsewhere e.g., line 34; 178).

--Line 190 – tests are not reliable, scores that come from them can be;
and reliability is a matter of degree not all-or-none

Author Response

Brain Sciences

Re:  Manuscript No. brainsci – 224795 – Original.docx

Thank you for your reviewer comments on this manuscript.  The manuscript continues to improve with your feedback and scholarly advice.

Comments from Reviewer 1:

Thank you for your comments regarding the concise overview the paper provides.  Thank you for the comments regarding the scope of information and literature selection. 

We have attempted to rely on primary sources in several areas.

Line 27 and 28:  We have removed the term “cognitive function” as it was redundant.

Line 33:  We have added the right parenthesis.

Line 39 and discussion on suicidality after concussion:  We agree with the reviewer’s concern regarding the hyperbole in the discussion regarding suicide after concussion.  However, we do feel the need to acknowledge this type of literature. 

Line 40:  We have removed the term “suicide” and inserted “affective disturbances”.  We feel this decreases the potential for alarm expressed by the reviewer.

In what used to be line 57 and is now line 62, we have removed “information processing tasks”, and substituted it with the statement “tasks that assess processing speed and attention such as reciting the months of the year backward”.

Line 96:  We have added the period.

Line 106:  We are back to the discussion of the potential of suicide in concussed patients.  We have simply added the reviewer’s suggested statement in line 117 “it is unknown if concussion increases the risk for suicide in any meaningful way”.

Line 35:  We have tried to keep our references below 40.

Line 190:  We have removed the term “reliable” from the manuscript. 

Thank you for helping to improve this manuscript. 

Reviewer 2 Report

This MUST be Labelled as a THOUGHT Piece

As a thought piece it has approached the process of evaluation of the client/patient in the office in a more thorough and systematic way to evaluate the effects and potential problems post concussion.  The acronym utilized is interesting but reflects a field side perspective as opposed to an in office perspective by its very name.(Minor issue).

There is no data presented as to how many individuals would fall into each of the 7 streams and what percentage would overlap 2 or more at least initially.

Further guidance as to how to direct care after this evaluation I'm sure is the next step but primary care practitioners would rather have this information packaged together with the assessment in an algorithm to help advance and streamline their practice.

It is a clearer article now from before, but to increase impact some of the frequency data for each stream might help cement the desired impact for the front line health care practitioner.

Sentence 44-45 needs correction:  The Berlin document has stated that "the" concussion diagnosis should not be "made", if another condition can explain the clinical findings.

The Numbering system is unusual

Don't number the introduction and then start with 1- for Cognitive function and number 1 through 7

Only one paediatric reference - line 97 {I am a paediatrician}

Line 152 "this_phenotype(34)."

Section 8 - discussion of Occular motor functionand the VOMS under the vestibular section which is fine but perhaps this needs to be discussed in section 3 Occular Motor first and then cross referenced later.

Author Response

Brain Sciences

Re:  Manuscript No. brainsci – 224795 – Original.docx

Comments from Reviewer 2:

Thank you for your comments regarding the thorough and systemic way the paper outlines the evaluation of concussion.  We acknowledge the sporting reference in the acronym, but we think this is reflective of the majority of literature being from the domain of sport-related concussion.  We are currently supporting research into non-sport-related concussion. 

We have attempted to indicate that the phenotypes overlap, and have done this in lines 48 to 52 in the revised manuscript.  We have also tried to provide percentage data when available.  We have included percentage data for oculomotor dysfunction, cardiovascular dysfunction and vestibular dysfunction. The oculomotor percentage is on line 76.  The cardiovascular percentages are on lines 179 and 180.  The vestibular percentages are on line 190.  We have tried to include the treatment options for individuals in the specific phenotypes and provide this in the Oculomotor section, the Affective section, the Cervical section, the Headache section and the Vestibular section.

The concerns with sentence 44 to 45 are essentially a direct quote from that paper.  We have put a reference to the paper in that section.  We do not use the term “made” but do review this important exclusionary clause in the Berlin document. 

We have changed the numbering system so that we simply now go through the seven different phenotypes and number them accordingly. 

We have added a pediatric reference at the end including vestibular patients.  References 3, 5, 8, and 20 evaluated a pediatric population. 

We have added the space at line 152. 

We have left the VOMS in the vestibular section.  We agree that there is substantial overlap between these two phenotypes.  They still are clinically distinct, however.  For example, optometrists are seeing many concussion patients, but do not routinely evaluate patients for vestibular pathology.

Thank you for your contributions to this manuscript. 

Round 2

Reviewer 2 Report

THANK YOU FOR THE REVISIONS